# TDP-43 Proteinopathy and Tauopathy: Do They Have Pathomechanistic Links?

**DOI:** 10.3390/ijms232415755

**Published:** 2022-12-12

**Authors:** Yuichi Riku, Mari Yoshida, Yasushi Iwasaki, Gen Sobue, Masahisa Katsuno, Shinsuke Ishigaki

**Affiliations:** 1Institute for Medical Science of Aging, Aichi Medical University, Nagakute 480-1195, Japan; 2Department of Neurology, Nagoya University Graduate School of Medicine, Nagoya 744-8550, Japan; 3Graduate School of Medicine, Aichi Medical University, Nagakute 480-1195, Japan; 4Department of Clinical Research Education, Nagoya University Graduate School of Medicine, Nagoya 744-8550, Japan; 5Molecular Neuroscience Research Center, Shiga University of Medical Science, Otsu 520-2192, Japan

**Keywords:** AD, ALS, CBD, FTLD, LATE, PART, PSP, TDP-43, SFPQ, tau

## Abstract

Transactivation response DNA binding protein 43 kDa (TDP-43) and tau are major pathological proteins of neurodegenerative disorders, of which neuronal and glial aggregates are pathological hallmarks. Interestingly, accumulating evidence from neuropathological studies has shown that comorbid TDP-43 pathology is observed in a subset of patients with tauopathies, and vice versa. The concomitant pathology often spreads in a disease-specific manner and has morphological characteristics in each primary disorder. The findings from translational studies have suggested that comorbid TDP-43 or tau pathology has clinical impacts and that the comorbid pathology is not a bystander, but a part of the disease process. Shared genetic risk factors or molecular abnormalities between TDP-43 proteinopathies and tauopathies, and direct interactions between TDP-43 and tau aggregates, have been reported. Further investigations to clarify the pathogenetic factors that are shared by a broad spectrum of neurodegenerative disorders will establish key therapeutic targets.

## 1. Introduction

Neuronal and glial aggregates of misfolded proteins, as well as systematic neuron loss and astrogliosis, are pathological hallmarks of neurodegenerative disorders. Transactivation response DNA binding protein 43 kDa (TDP-43/TARDBP) is a pathological protein observed in amyotrophic lateral sclerosis (ALS) and frontotemporal lobar degeneration (FTLD) [1,2,3], and a subset of aged people also show TDP-43-related pathology [4]. TDP-43 related ALS and FTLD (ALS-TDP and FTLD-TDP) and aging-related TDP-43 pathology can collectively be termed TDP-43 proteinopathies. Tau protein is associated with Alzheimer disease (AD), Pick disease (PiD), progressive supranuclear palsy (PSP), corticobasal degeneration (CBD), argyrophilic grain disease (AGD), and globular glial tauopathy (GGT) [5,6], and aged people often have neuronal or glial tau pathology [7,8]; these conditions are termed tauopathies. The peptide or protein components, morphology, and spatial distributions of the aggregates, in association with antemortem neurological assessments, are fundamental in the final diagnosis of neurodegenerative disorders.

Numerous basic studies reproducing pathological aggregates of TDP-43 and tau have revealed the neurotoxicity of the aggregates or loss of physiological functions of these proteins, although it is often controversial which of them is the more prominent pathway in patients [9]. By contrast, upstream pathways resulting in TDP-43 or tau pathology remain to be elucidated, and it is unclear why neurodegenerative disorders affect certain individuals. Genetic approaches have clarified the pathogenic gene mutations responsible for inherited tauopathies and TDP-43 proteinopathies, and biochemical studies have addressed the mutation-derived misfolding or dysfunction of these proteins. However, for sporadic patients with TDP-43 proteinopathies or tauopathies, the pathways upstream of protein aggregates are not fully understood, and most patients with neurodegenerative disorders are sporadic. In other words, it is presumed that the entire pathogeneses of sporadic neurodegenerative disorders can hardly be explained by the alteration of a single gene or a molecule.

Interestingly, patient autopsies have revealed comorbid TDP-43 pathology in tauopathy patients, and vice versa. For example, a subset of AD patients showed TDP-43 pathology in the limbic system [10,11]. Comorbid TDP-43 pathology in AD brains shows characteristic patterns of spatial spreading and morphological features that are different from those in ALS/FTLD-TDP. TDP-43 pathology positively correlates with the severity of brain atrophy and cognitive impairment in patients with AD neuropathological changes [10,11]. Hence, it can be hypothesized that the comorbid pathology is a significant part of the pathogenesis in each neurodegenerative disease. Moreover, autopsy studies from various spectra of neurological disorders have clarified that both TDP-43 and tau pathologies can be simultaneously observed in patients with certain metabolic disorders, inflammatory disorders, or neoplasms (Table). These facts indicate that the mechanisms forming TDP-43 pathology and tau pathology are partially overlapped. Basic and translational research has also addressed common pathogenic factors between TDP-43 proteinopathies and tauopathies, or the direct interaction between aggregates of TDP-43 and tau. Research of pathomechanistic links is a reasonable way to clarify key molecules as therapeutic targets for neurodegenerative disorders that are untreatable, multifactorial, and polygenetic. In this review, we discuss the links between TDP-43 proteinopathies and tauopathies at neuropathological (patient), genetical, and basic (model) levels.

## 2. Neuropathology of TDP-43 Proteinopathies

TDP-43 is an RNA-binding protein that is coded on the chromosome 1p. Systemic organs, including the central nervous system, pancreas, and spleen, abundantly express TDP-43 [1]. TDP-43 contains two RNA-recognition motifs (RRMs), a nuclear localization signal (NLS), and a nuclear export signal (NES) that are critical for intracellular localization. The C-terminal side of TDP-43 contains a prion-like domain (glycine-rich domain) [9,12]. This prion-like domain contains abundant polar amino acids, with sparse charged acids, and is critical for the phase separation of TDP-43 [12]. In 2006, TDP-43 was identified as a main component of the ubiquitylated aggregates observed in the neurons of ALS and FTLD patients [2,3]. In ALS patients, the affected motor neurons exhibit a mislocalization of nuclear TDP-43 and cytoplasmic aggregates (Figure 1); the term “TDP-43 pathology” is usually defined as a combination of the disappearance from the nucleus and cytoplasmic aggregation of TDP-43, as in this article. More than 90% of sporadic ALS patients were reported to demonstrate TDP-43 pathology [13]. 

Another pathological finding of ALS-TDP is Bunina bodies. Bunina bodies are highly eosinophilic granules, mainly found within the motor neuron cytoplasm, and which show immunoreactivity for cystatin-C and transferrin, but rarely for TDP-43 [15]. Although Bunina bodies are not found among non-TDP-43-related ALS/FTLD, the mechanisms underlying the formation of this mysterious inclusion are unknown.

TDP-43 is also known as a pathological protein of FTLD; each of TDP-43 and tau accounts for nearly 50% of total FTLD patients. The term “FTLD” encompasses the pathological entity corresponding to a clinical term “frontotemporal dementia (FTD)”. FTLD-TDP is characterized by a prominent TDP-43 pathology in the frontotemporal cortices, and the limbic system, hippocampus, neostriatum, and substantia nigra are also vulnerable [16,17,18]. Pathological criteria subclassify FTLD-TDP into four groups: types A, B, C, and D [19] (Figure 1). Type A shows dense TDP-43 aggregates in the neuronal cytoplasm and short aggregates in neurites prominent in the superficial layers of the cerebral cortices; this type is clinically associated with behavior-variant frontotemporal dementia (bv-FTD) or progressive non-fluent aphasia. Type B demonstrates TDP-43 pathology across all cortical layers, and neuritic aggregates are rare; this type is associated with bv-FTD and FTD with motor neuron disease (FTD-MND). Type C is characterized by neuritic TDP-43 aggregates that are longer and thicker than those in type A, while cytoplasmic aggregates are rare in this type. Bv-FTD and semantic dementia are clinical characteristics of type C. Type D shows lentiform intranuclear inclusions of TDP-43 and is associated with mutation of *valocin-containing-protein* (*VCP*) gene [20]. The clinical phenotype of type D is known as a combined syndrome of inclusion body myopathy, Paget’s disease, and frontotemporal dementia (IBMPFD), but recent studies have revealed that the clinical phenotypes of *VCP* mutation are quite diverse among families or even within a single family [21].

TDP-43 pathology is also common in familial ALS and FTLD. The most frequent gene mutation among familial ALS/FTLD-TDP is hxanucleotide (intronic GGGGCC) expansion of *C9orf72* [22,23]; approximately 40% of familial ALS patients have this mutation, and the mutation can manifest as any of the ALS, FTD-MND, and FTD phenotypes [13]. Another common mutation of familial FTLD-TDP is *progranulin (GRN)*, which is clinically associated with the pure FTD form with an absence of motor neuron degeneration [24,25]. The *TDP-43* mutation is also found in familial ALS; the *TDP-43* mutations typically manifest as ALS [26,27] and are occasionally associated with FTLD phenotype [28]. The known mutation sites in *TDP-43* are mostly located on the C-terminal side, including the prion-like domain.

ALS-TDP and FTLD-TDP seem to have a close relationship. It is well known by neurologists that ALS sometimes co-occurs with FTD in patients, which has been termed FTD-MND or ALS with dementia. Neuropathological observations of autopsied patients have also suggested a relationship between ALS and FTLD. Studies have reported that TDP-43 pathology extends across multiple systems in the brains of ALS and FTLD patients, and the distributions of the lesions often overlap between these diseases. For example, the prefrontal cortices, primary motor cortex, limbic system, hippocampus, parahippocampal gyrus, and neostriatum are involved in, not only FTLD-TDP, but also a subset of ALS-TDP [29]. Furthermore, a study revealed that most patients with FTLD-TDP had TDP-43 pathology and Bunina bodies in the lower motor neurons of the spinal cord and brain stem, even when they lacked lower motor neuron symptoms and signs [30]. It has also been reported that a subset of ALS patients, who were on long-term artificial ventilation, have a broadly extended TDP-43 pathology beyond the motor neuron system [31], although it is controversial whether such a phenotype and FTLD-TDP are identical. The clinical and neuropathological overlap between ALS-TDP and FTLD-TDP may support the concept of TDP-43 proteinopathies, where these two disorders are included in a continuum of the disease spectrum.

TDP-43 pathology can be found in some aged people. Aging-related TDP-43 pathology is typically prominent in the limbic system, hence being termed limbic-predominant age-related TDP-43 encephalopathy (LATE); a recent neuropathological guideline defines LATE stages as stage 1 (amygdala only), 2 (spreading to the hippocampus), and 3 (spreading to the middle frontal gyrus) [4]. A study reported that LATE and the TDP-43 pathology of ALS/FTLD-TDP can be clearly distinguished by neuropathologists, with high sensitivity and specificity [32]. This mainly arises from the broader distribution and higher density of cortical TDP-43 pathology in ALS/FTLD-TDP than those in LATE. Moreover, ages at death are higher in LATE subjects than in ALS/FTLD-TDP patients. It is an interesting point whether LATE is a nonharmful bystander or impairs cognitive functions. A retrospective, clinicopathological study reported that the cognitive function of patients with early AD pathology was more impaired when they had LATE than when they did not [33].

Basic research has accumulated evidence that the cytoplasmic inclusions of TDP-43 are neurotoxic. Neuronal death or axonal dysfunction has been reported using models with TDP-43 overexpression [34], transfection of pathogenic mutations of TDP-43 [35,36], or cytoplasmic mislocalization of TDP-43 with mutated NLSs [37]. Moreover, TDP-43-aggregates have a seeding property, for further development of the aggregation; cell-to-cell transmission of injected TDP-43 aggregates have been reported in neuron cell lines [38] and in mice expressing human TDP-43 with a mutated NLS [39]. Findings from seeding experiments suggest a “prion hypothesis”, in which insoluble, neurotoxic protein aggregates are hypothesized to spread in a cell-to-cell manner. In autopsy-based research, it has been indicated that the frequency of TDP-43 pathology in ALS patients shows a corticofugal descending gradient through the anatomic neural systems; it starts from the primary motor cortex and then spreads to postsynaptic neurons, including the lower motor neurons, striatal neurons, and those in other frontal and temporal areas. This fact is fundamental concept of the “propagation hypothesis”, which speculates that protein aggregates spread in a neuron-to-neuron manner via trans-synaptic transfer [40].

By contrast, the loss of nuclear TDP-43 has also been suggested to be a TDP-43-mediated pathomechanism. Transgenic mice expressing human TDP-43 with a mutated NLS displayed neuronal loss and tract degeneration in association with the downregulation of endogenous nuclear TDP-43, but with sparse cytoplasmic inclusions; loss of nuclear TDP-43, rather than cytoplasmic inclusions, was correlated with neuronal dysfunctions in this study [41]. Conditional depletion or knockout models of TDP-43 have shown ALS-like clinical phenotypes [42] or cellular dysfunctions, including deficits in DNA repair [43], an alteration of TDP-43-transcriptome resulting in synaptic impairment [44], a loss of splicing repressor function [45], and astrocytic activation [46].

## 3. Neuropathology of Tauopathies

Tauopathies are defined by neuronal or glial aggregates of hyperphosphorylated tau and are subclassified into 3R-tauopathies and 4R-tauopathies, according to the prominent isoforms of tau aggregates [5] (Figure 2). Tau is distributed in nervous and muscular tissues and is most abundant in the neurons of the central nervous system. *Microtubule-associated protein tau* (*MAPT*) gene encodes tau protein, and the alternative splicing of exons 2, 3, and 10 in *MAPT* generates six isoforms. These isoforms are largely subclassified into two isoform groups: 3-repeat (3R) and 4-repeat (4R) tau. MAPT exon 10 is located in microtubule-binding repeats, and the 4R isoform contains exon 10, whereas the 3R isoform lacks this. The microtubule-binding repeats physiologically interact with the C-terminal end of tau, forming a folding structure, even in a soluble state [47]. The repeats contain a paired-helical filament (PHF) motif, the hexapeptide motif (VQIVYK), that is a critical site for misfolding and has a β-sheet structure [48]. Cryo-electron microscopy has revealed that tau-filament cores in AD patients are made of two identical protofilaments comprising residues V306–F378 of tau protein [49], whereas those in PiD patients consist of residues K254–F378 of 3R-tau [50]; these studies suggest that the molecular structures of misfolded tau differ among tauopathies. In vivo studies revealed that peripherally or intracerebrally injected misfolded tau functions as a seed for tau aggregation, resulting in a spread to different brain regions distant from the injection sites [51,52]. Misfolded tau also binds to intraneuronal organelles, including mitochondria [53] and presynaptic vesicles [54].

Mutations of the *MAPT* gene were identified as a cause of chromosome-17-related familial frontotemporal dementia with parkinsonism (FTDP-17) [55], although another causative gene of FTDP-17 is *GRN*. Approximately 50 mutations have been identified, which can be associated with diverse 3R-tauopathy and 4R-tauopathy phenotypes, according to the mutation site [56]. Although the pathways between *MAPT* mutations and tau aggregation have not been fully clarified, most mutation sites have been found in exons 9–12 and adjacent introns and associated with altered microtubule assembly abilities [56]. It has also been reported that the P301L mutation, a common mutation site in the microtubule-binding domain, enhanced the seeded recruitment of endogenous tau in cultured cells [57].

### 3.1. 3R-Tauopathy (PiD)

Patients with PiD show prominent brain atrophy and neuron loss in the prefrontal area and the bottom sides of the temporal lobes. The primary motor cortex, the most posterior region of the frontal lobes, is less involved than the prefrontal area, representing a contrast with other FTLDs. A characteristic neuropathological finding of PiD is Pick bodies, which appear as spherical inclusions within the neuronal cytoplasm, typically nearby the apical dendrites. Pick bodies are aggregates of 3R-tau and strongly labeled with silver staining methods. Neurons having Pick bodies also show diffuse immunoreactivity for hyperphosphorylated tau in the cytoplasm. The hippocampal granule cells often demonstrate numerous Pick bodies in association with severe neuron loss. Ballooned neurons that display swelling of cytoplasm with deviated nuclei are also abundant in PiD.

### 3.2. 4R-Tauopathies (PSP, CBD, AGD, and GGT)

PSP systematically involves the globus pallidus, subthalamic nucleus, and tegmentum of the midbrain and pons, in combination with impairment of the cerebellar efferent system, including the dentate nucleus and upper cerebellar peduncle. Lesions show globose neurofibrillary tangles in the neuronal cytoplasm and tufted astrocytes that are labelled using anti-4R-tau immunohistochemistry and silver staining methods. White matters display coiled bodies that are tau aggregates in oligodendrocytes.

CBD grossly shows para-sylvian atrophy, which is often asymmetrical, in association with atrophy of the basal ganglia. The most striking microscopic finding is dense and numerous neuropil threads that are immunopositive for hyperphosphorylated tau in the deep cortical layers and white matters. Neuronal cytoplasm shows pretangles that are more fuzzy and more diffusely occupy the cytoplasm than NFTs. Ballooned neurons are often observed. Astrocytic plaques, tau aggregations within the distal portions of astrocytic processes, are characteristic glial inclusions of CBD; this is in contrast with the tufted astrocytes of PSP that are prominent in cytoplasm and proximal processes.

AGD is characterized by 4R-tau-immunopositive and argyrophilic granules in the neuropil. AGD preferentially involves the anterior portion of parahippocampal gyrus and subiculum, showing atrophy of the medial temporal lobes. In a small subset of patients, the argyrophilic grains are extended to the limbic systems, basal ganglia, and frontal cortices. Ballooned neurons with tau-immunopositivity are often observed in the parahippocampal gyrus and hippocampal granule cells.

GGT is a recently-identified 4R-tauopathy. Although several subtypes have been published for this disease, the typical findings are thick and globular aggregates of 4R-tau in oligodendrocytes, which are clearly distinguishable from the coiled bodies in other tauopathies.

### 3.3. 3R and 4R-Tauopathy (AD)

Neuropathological findings of AD comprise neurofibrillary tangles (NFTs) and senile plaques. NFTs are argyrophilic flame-like structures that are composed of 3R-tau and 4R-tau aggregations; it has been reported that 4R-tau is prominent during the early process of tau aggregation, whereas 3R-tau is increased during the maturation of NFTs in AD [58]. It is well known that the spreading of NFTs has a hierarchy, starting from the entorhinal cortex, subiculum, and CA1, and then extending to the neocortex during advanced disease phases [59]. Senile plaques are argyrophilic, Aβ-amyloid deposits that are constituted by a neuritic core and surrounding deposits. Anti-Aβ-amyloid immunohistochemistry also recognizes diffuse plaques, which lack neuritic cores and definite silver staining. Deposition of Aβ-amyloid has a certain spreading pattern that starts from the neocortex, and then moves to the hippocampus, followed by the basal ganglia, brainstem, and cerebellum [60].

### 3.4. Aging-Related Tauopathies

A subset of neurologically healthy people show local deposition of hyperphosphorylated tau in the brain. Age-related tau aggregation includes primary age-related tauopathy (PART) [7] and aging-related tau astrogliopathy (ARTAG) [8]. PART is a combined 3R and 4R tauopathy. The distribution, progression, and morphology of tau aggregates in PART are similar to those in early AD, but Aβ-amyloid deposition is absent or sparse. It is controversial whether PART can be clearly distinguished from early pathological changes of AD [61]. It has been reported that PART can develop in the absence of a senile plaque and manifest as dementia; namely, senile dementia of the neurofibrillary tangle type (SD-NFT) or tangle-only dementia [62,63]. ARTAG is a 4R tauopathy. ARTAG shows characteristic tau aggregates in the astrocytic cytoplasm, including granular, fuzzy, and thorn-shaped aggregates, which are typically prominent in the subependymal or subpial regions [8].

## 4. Comorbid TDP-43 Pathology in Tauopathies and Comorbid Tau Pathology in TDP-43 Proteinopathies

Postmortem studies have reported that TDP-43 aggregates can be observed in brain tissue under non-ALS/FTLD disorders, including AD [64,65] (Figure 3), PSP [66,67] (Figure 3), CBD [68,69] (Figure 3), Lewy body diseases [70,71,72], hippocampal sclerosis [73], post-traumatic chronic encephalopathy [74], and brain tumors [75]. Rarely, concurrence of ALS/FTLD-TDP together with PSP, CBD, or AD has been reported [76,77]. However, “comorbid pathology” is clearly distinguished from such “concomitant disorders” by the limited extension of TDP-43 pathology, lack of clinical findings diagnostic of ALS/FTLD-TDP, and a much higher prevalence than that expected for the incidental concurrence of two neurodegenerative disorders.

Various neuropathological studies have emphasized that the comorbid TDP-43 pathology in tauopathies, including AD, PSP, and CBD, has disease-specific characteristics. A study revealed that TDP-43 pathology was observed in 195 (57%) out of 342 AD patients [10]. TDP-43 pathology in the AD brain is morphologically similar to LATE and also spreads in a LATE-like hierarchy; it starts from the amygdala (stage 1) and then moves to the entorhinal cortex and subiculum (stage 2); to the dentate gyrus of the hippocampus and occipitotemporal cortex (stage 3); to the insular cortex, ventral striatum, basal forebrain, and inferior temporal cortex (stage 4); to the substantia nigra, inferior olive, and midbrain tectum (stage 5); and finally to the basal ganglia and middle frontal cortex (stage 6) [10]. A later study revealed that comorbid TDP-43 pathology is positively correlated with the severity of brain atrophy in AD patients [11]. These facts indicate that TDP-43 pathology develops alongside the progression of AD pathogenesis. Additionally, an autopsy-based study reported different biochemical properties between TDP-43 aggregates of AD patients and those of FTLD-TDP patients; neuronal cytoplasmic or neuritic TDP-43 aggregates in AD patients demonstrated a varied immunoreactivity for full-length TDP-43, phosphorylated TDP-43 s402/403, or phosphorylated TDP-43 s409/410 among individuals, whereas TDP-43 aggregates in FTLD-TDP patients consistently showed immunoreactivity for all molecular forms of TDP-43 [78].

The characteristic TDP-43 pathology has also been found in 4R-tauopathies, including PSP and CBD. It was reported that 47 out of 945 PSP patients (5%) showed TDP-43 aggregates in the hippocampus [66]. A study revealed that five of 19 patients (26%) with PSP showed TDP-43 aggregates in the limbic area [67]. Another study of CBD patients revealed that 84 out of 187 patients (45%) had TDP-43 aggregates in the midbrain tegmentum or other brain regions [68]. A study of 45 CBD patients and 108 PSP patients found that 14% of PSP and 33% of CBD patients had TDP-43 pathology in the brain [79]. Spatial analyses indicated that TDP-43 pathology in PSP and CBD patients is clearly distinguished from that in LATE. It was reported that comorbid TDP-43 pathology is prominent in not only the limbic system but also the substantia nigra, neostriatum, and globus pallidus among PSP patients [66,67]. A study of CBD patients found that comorbid TDP-43 pathology was prominent in the midbrain tegmentum, followed by the subthalamic nucleus [68]. Some of the above studies addressed the clinical impacts of comorbid TDP-43 pathology. For example, TDP-43 pathology in the hippocampus of PSP patients was associated with hippocampal sclerosis [67]. TDP-43 pathology in the midbrain tectum of CBD patients was correlated with the presence of downward gaze palsy [68]. Recently, we reported that 10 of 26 PSP patients (38%) and seven of 12 CBD patients (58%) showed TDP-43 pathology in spinal cord motor neurons [80]. The motor neuron TDP-43 pathology in PSP and CBD was morphologically similar to those in ALS/FTLD-TDP, and aggregated TDP-43 was truncated at the C-terminus, showing the same blotting patterns as ALS/FTLD-TDP. The severity and prevalence of lower motor neuron TDP-43 pathology were positively correlated with those of 4R-tau pathology but not with aging. The burden of reactive microglia was positively correlated with the severity of TDP-43 pathology in the spinal cord of PSP and CBD patients, although lower motor neuron symptoms had not been detected in the studied patients [80]. The motor neuron TDP-43 pathology in PSP and CBD patients also showed a difference from that in ALS/FTLD-TDP; the TDP-43 pathology was consistently prominent in the cervical cord, which forms a great contrast with ALS/FTLD-TDP, which involves broad segments and brainstem motor neurons. Definite neuron loss and classic Bunina bodies were absent in PSP and CBD patients having TDP-43 pathology [80].

**Figure 3 ijms-23-15755-f003:**
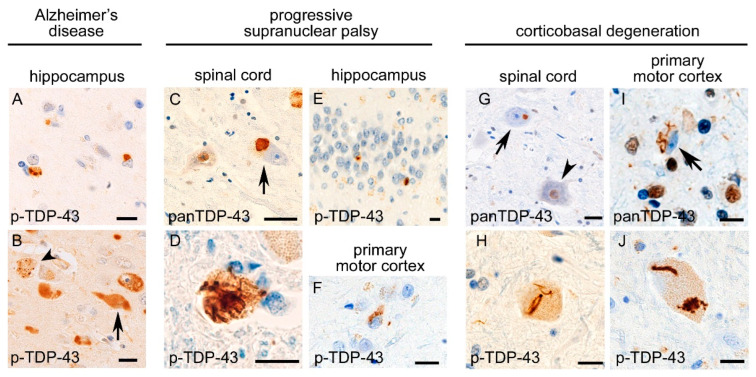
Comorbid TDP-43 pathology in tauopathies. TDP-43 aggregates of Alzheimer’s disease patients are observed prominently in the limbic system, of which morphological characteristics and distributions are similar to those of LATE (**A**). Anti-phosphorylated TDP-43 immunohistochemistry also shows NFT-like aggregates (**B**, arrow) and recognizes granulovacuolar degeneration (**B**, arrowhead). In progressive supranuclear palsy, the spinal cord motor neuron demonstrates cytoplasmic aggregates and nuclear mislocalization of TDP-43 (**C**, arrow, and **D**). TDP-43 pathology is also observed in the hippocampal granule cells (**E**) and primary motor cortex (**F**) [80]. In corticobasal degeneration, spinal cord motor neuron (**G**, arrow, and **H**) and small pyramidal neuron and Betz cell in the primary motor cortex (**I**,**J**) show TDP-43 pathology. (**A**,**B**,**D**–**F**,**H**,**J**) Anti-phosphorylated TDP-43 and (**C**,**G**,**I**) anti-panTDP-43 immunohistochemistry. (**A**) amygdala, (**B**) hippocampus. Bars: 10 μm.

Conversely, the comorbidity of tau aggregation has been reported in ALS-TDP and FTLD-TDP. A study reported that 24 (59%) of 41 ALS patients, seven (44%) of 16 FTD-MND patients, and 10 (43%) of 23 FTD patients showed tau pathology [81]. Tau pathology in these patients had the characteristics of AD or PART, but a subset of patients showed novel tau pathology characterized by a fine granular or more clumped aggregation, without neurofibrillary tangle structures or neuropil threads in the frontal cortex, which were specifically detected using a pThr175 antibody [81]. Another study addressing 10 ALS-TDP patients reported that tau aggregation was observed in the entorhinal cortex of all assayed patients, and ALS patients with cognitive impairment showed a broader extension of neuronal and glial tau pathology than those without cognitive impairment [82]. It was reported that phosphorylated tau epitope was increased in the nucleus and cytoplasm of lower motor neurons among ALS patients, whereas the immunoreactivity was almost negative in the lower motor neurons of controls [83]. *C9orf72* hexanucleotide expansion was associated with more abundant tau aggregates in the limbic system than *GRN* mutation or sporadic ALS/FTLD-TDP [84]. These study results suggest the possibility that ALS/FTLD-TDP have a risk for disease-specific tau pathology. However, contradictory results have also been reported. A study addressing 33 patients with tauopathies and 45 patients with TDP-43 proteinopathies revealed that the extension and prevalence of comorbid TDP-43 pathology or tau pathology in each disorder group were within the range of aging pathologies [85].

Several studies have addressed the colocalization of TDP-43 and tau aggregates in tauopathies. A study of the lower motor neuron of PSP and CBD patients reported that TDP-43 pathology was fundamentally observed in the neurons without tau aggregates, and colocalization of TDP-43 aggregates with tau aggregates were rare, even when they were present within the same neuronal cytoplasm [80]. Another study revealed that tau-immunopositive astrocytic plaque and neuropil threads were partially immunolabeled with TDP-43, but the fluorescent signals of tau and TDP-43 were not completely overlapped [69]. A study of AD patients reported that aggregates of TDP-43 and hyperphosphorylated tau often coexisted within the same neuronal soma of amygdala, hippocampus, and medial temporal cortices, but were rarely superimposed [72]. A recent study in an older population (88.4 years on average) addressed non-FTLD-related TDP-43 aggregates in the hippocampal sections, and less than 25% of TDP-43 aggregates were colocalized with tau in the hippocampal granule neurons [86]. Overall, TDP-43 aggregates often seem to be located at certain distances from tau aggregates in PSP, CBD, AD, and aged people; there may be parallel, not only secondary co-aggregation, pathways of TDP-43 pathology alongside the development of tau pathology.

## 5. Granulovacuolar Degeneration: A Unique Pathology Involving TDP-43 and Tau

Granulovacuolar degeneration (GVD) is defined as the presence of clusters of slightly basophilic granules within fused vacuoles, which are typically observed in the cytoplasm of pyramidal neurons [87] (Figure 4). Granules of GVD are often immunoreactive for hyperphosphorylated tau and phosphorylated TDP-43. Although anti-full-length TDP-43 immunohistochemistry also labels GVD granules, its immunostaining is much weaker than that of phosphorylated TDP-43, and intranuclear TDP-43 is usually spared in GVD-containing neurons. In addition, more than 30 epitopes have been identified in GVD samples, including autophagosome/endosomal proteins, lysosomal proteins, ubiquitin, and necrosomal proteins [87]. It has been reported that GVD resembles autophagic organelles, suggesting that GVD reflects a perturbed maturation of the autophagosome-lysosome cascade [88]. GVD is frequently observed in AD patients and is not rare among patients with other tauopathies and cognitively healthy older populations [87,89]. The spread of GVD shows a clear hierarchy, starting from the hippocampal pyramidal neurons (particularly of CA1 and subiculum) and then moving to the entorhinal cortex, amygdala, hypothalamus, and cerebral neocortices in decreasing frequency [90]. GVD-containing neurons often show cytoplasmic aggregation of hyperphosphorylated tau, and GVD is hypothesized to be a trigger of neuronal tau pathology [87]. Several studies have also reported a relationship between GVD and TDP-43 proteinopathies. A postmortem study revealed a higher prevalence of GVD in ALS/FTLD-TDP patients with *C9orf72* hexanucleotide expansion than in nonmutated ALS/FTLD-TDP patients or age-matched controls [91]. C9orf72 is known as a regulator of the autophagosome-lysosomal pathway [92]. It is interesting that GVD granules abundantly contain autophagosome-lysosome-related epitopes, even though *C9orf72* hexanucleotide expansion and its haploinsufficiency are not the same. In particular, GVD granules intensely express charged multivesicular body protein 2B (CHMP2B), which is a marker of endosomal vesicles. Mutations of the *CHMP2B* gene are known to cause familial FTD and ALS [93,94], and mutation carriers have TDP-43 pathology [94]. A later study assessing 30 patients with ALS/FTLD-TDP, including those with and without *C9orf72* hexanucleotide expansion, found that GVD expressing necrosome markers was spread with the extension of TDP-43 aggregates, as well as tau aggregates [95]. Taken together, these findings indicate that GVD may be involved in the pathogenesis of, not only tauopathies, but also TDP-43 proteinopathies. However, it remains unsolved why GVD is absent in the lower motor neurons that are preferentially involved in ALS/FTLD-TDP-43.

## 6. Do TDP-43 Proteinopathy and Tauopathy Have Mechanical Links?

Table 1 summarizes the diseases or conditions in which both TDP-43 and tau aggregations can be simultaneously observed [96,97,98,99,100,101,102,103,104,105,106,107,108,109,110,111,112,113,114]. These conditions include aging, tumors, inflammatory processes, and gene mutations, as well as neurodegenerative disorders. It is interesting that the pathogeneses underlying them are often related to the autophagosome-lysosome system, because this system is critical for the transport and metabolism of p62-tagged ubiquitylated proteins in lysosome [115] and also leads to recruitment of exosome, which is important for extra-cellular transfer of degraded proteins [116]. Optineurin (OPTN), progranulin, C9orf72, CHMP2B, transmembrane protein 106B (TMEM106B), VCP, and mTOR-signaling pathway are key molecules participating in the autophagosome-lysosome system [117], and the immunohistochemical properties of inclusion bodies observed in IBM muscles are highly similar to those of GVD [100]. Basic research has partially supported that alterations of these “key molecules” can reproduce both TDP-43 and tau aggregations. Progranulin deficiency leads to mislocalization [118] or intranuclear aggregates of TDP-43 [119] in primary neuronal cultures. It has also been reported that haploinsufficiency of *grn* increased hyperphosphorylated tau in P301L-Tg mice [120]. Furthermore, a study revealed that double knockout of *grn* and *tmem106b* in mice exhibited more severe TDP-43 aggregates than single knockout of *grn* [121]. An in vivo model of *c9orf72* insufficiency reproduced the aggregation of p62-immunopositive protein aggregates via an altered endosomal trafficking ability [92]. Deregulation of mTOR signaling is known to inhibit autophagy [122], and in vitro studies showed that activation of mTOR signaling results in enhanced tau hyperphosphorylation [123]. Moreover, a study using FTLD model mice showed that inhibition of mTOR by rapamycin decreased cytoplasmic TDP-43 inclusions, in association with increased autophagic markers [124]. Thus, alterations to the autophagy/lysosome system-related molecules or genes shown above seem to be critical to both tau and TDP-43 pathologies at the level of basic research.

In vivo studies have also indicated a tau-seeding activity is demonstrated by misfolded TDP-43. A study found that human mutant-TDP-43-knock-in mice showed alterations in *mapt* exon 2/3 splicing, which are critical regions for the differentiation of tau protein isoforms. *Mapt* splice variant expression (1N/0N) was significantly altered in *tardbp* Q331K homozygous mice compared to heterozygous or wild-type mice. The study concluded that TDP-43 binds to an intronic sequence upstream of *mapt* exon 2 [126]. It has also been reported that subjects with human mutated *TDP-43* M337V and *MAPT* T175D co-expression demonstrated increased aggregation of hyperphosphorylated tau in the hippocampus, when compared to subjects expressing wild-type tau, suggesting a synergistic effect of these human-derived aggregates toward tau pathology [127]. An in vitro experiment showed that patient brain-derived TDP-43 oligomers demonstrated a seeding activity for tau aggregates in wild-type tau- and P301L-mutated tau-inducible cells [128]. Data from these studies suggested that misfolded or oligomeric TDP-43 may have a nature that facilitates aggregation of tau.

Patient-based, genetic studies have emphasized the particular relationship between ALS/FTLD-TDP and 4R-tauopathies. Genome-wide association studies revealed selective genetic overlaps among ALS, FTLD, PSP, and CBD, which were absent in patients with AD and Parkinson’s disease and healthy controls [129,130]. ALS/FTLD-TDP patients harboring the *TMEM106B* rs1990622 A/A genotype, a known risk allele for ALS/FTLD-TDP, often show comorbid neuroglial aggregates of 4R-tau [125]. Intermediate GGGGCC repeat length (17–29) of *C9orf72* is associated with risk of CBD [131]. A later study, however, reported that there were no significant differences in *TMEM106B* genotypes or hexanucleotide lengths of *C9orf72* between PSP/CBD patients with and without TDP-43 pathology [80]; this may suggest that comorbidity of TDP-43 pathology is not derived by only a single genetic factor.

Translational research also supports a mechanistic link between ALS/FTLD-TDP and 4R-tauopathies (Figure 5). We have suggested that the altered interaction of two RNA-binding proteins, splicing factor proline/glutamine rich (SFPQ) and fused-in-sarcoma (FUS), is a candidate pathway shared by these two major disease groups. The interaction of SFPQ and FUS regulates *mapt* exon 10 splicing in mice expressing human-strain 3R- and 4R-tau isoforms, and impaired interaction of these molecules results in a 4R-tau-dominant condition [132]. It is important that FUS and SFPQ are also related to ALS pathogenesis. The mislocalization of SFPQ from motor neuron nuclei and SFPQ accumulation within TDP-43 aggregates have been reported as pathological hallmarks of ALS-TDP [133,134], and FUS is a major pathological protein of TDP-43-negative ALS [135,136]. Supporting this, postmortem studies revealed that the impaired interaction between SFPQ and FUS was a pathological change shared by the frontal, hippocampal, and spinal cord motor neurons among FTLD/ALS-TDP, PSP, and CBD patients; this finding was not observed in patients with AD and PiD and age-matched controls [80,137]; hence, impairment of SFPQ/FUS interaction has a disease specificity for ALS/FTLD-TDP and 4R-tauopathies. SFPQ, FUS, and TDP-43 are physiologically components of the para-speckle bodies in the neuronal nucleus. Para-speckle bodies are a complex of RNA-binding proteins and nuclear paraspeckle assembly transcript 1 (NEAT1), which is a V-shaped non-coding RNA running into the “core” structure and returning to the “shell” structure (Figure 1) [14]. SFPQ and FUS are components of the core structure in para-speckle body, whereas TDP-43 is a protein of the shell structure. Para-speckle bodies may be important scaffolds for these RNA-binding proteins in the nucleus, although the functions of these are not fully understood. It has been reported that other components of para-speckle bodies, non-POU domain-containing octamer-binding protein (NONO) and paraspeckle component 1 (PSPC1), are intactly localized within para-speckle bodies in ALS-TDP [138], in contrast to SFPQ, FUS, and TDP-43. Although the reason for this selectivity is unclear, accumulating data have shown that the prion-like domains of RNA-binding proteins are critical for building para-speckle bodies [14,139]. It is possible that degradation (misfolding or truncation) of the prion-like domain, of which TDP-43 and FUS are prominently involved by the aggregation and mutation, is responsible for the instability of components in para-speckle bodies.

Postmortem studies have suggested neuropathological overlapping between AD and LATE (Figure 5). As discussed in chapter 4, some studies have emphasized that TDP-43 pathology in the AD brain typically demonstrates LATE-like morphology and spatial distribution and may develop along with the disease progression of AD. However, it is not straightforward to discuss mechanistic links between AD and LATE. Neuropathological changes in AD comprise 3R-tau pathology, 4R-tau pathology, and Aβ-amyloid deposition. Hence, we cannot avoid complex discussions about the relationships between these three players and TDP-43. Recently, a postmortem study addressed positive correlations in severities between LATE and Aβ amyloid deposition, and between LATE and NFT [140]. The study, using a large case series from multiple centers, revealed that 54.9% of subjects with high grade neuritic amyloid plaques had comorbid LATE, whereas 27.0% of subjects with no neuritic amyloid plaques had LATE [140]. A sub-analysis on individuals lacking amyloid plaques revealed that the brains with LATE had relatively more severe PART than those without [140]. A genetical study of individuals with pathological changes of AD revealed that the apolipoprotein Eε4 allele, a known risk allele of Aβ-amyloid deposits, was strongly associated with positivity of TDP-43 pathology in the brain [141]. Thus, the mechanistic links between LATE and AD are becoming clearer. By contrast, the links between ALS/FTLD-TDP and AD remain controversial. A recent case control study reported that pathogenic mutation of *GRN*, a causative mutation of FTLD-TDP, was positively correlated with more severe tau aggregation in patients having AD pathology but not with Aβ-amyloid deposition [142].

## 7. Conclusions

Autopsy-based research has revealed that comorbid pathology often has a disease-specific manner, in terms of the biochemical properties, morphological characteristics, and spatial distributions of aggregates, and has an impact for clinical phenotypes. Genetic studies have identified overlapping genetic risk factors between LATE and AD or between ALS/FTLD-TDP and 4R-tauopathies. These facts indicate that comorbid pathology is not an incidental bystander, but a part of the disease pathogenesis. It will be important to determine when a TDP-43 or tau pathology is comorbid during disease pathogenesis; antemortem studies using functional neuroimaging targeting aggregated proteins will be useful in the future.

Basic research findings have suggested that the molecular pathways are partially overlapped between TDP-43 proteinopathies and tauopathies. In vivo studies have revealed that aggregated TDP-43 altered the splicing of tau or exacerbated tau aggregation. SFPQ, FUS, and TDP-43 were suggested to be commonly be involved by pathogeneses among ALS/FTLD-TDP, PSP, and CBD. Moreover, perturbation of the autophagosome-lysosome system-related molecules has been reported in both TDP-43 proteinopathy and tauopathy models. However, it currently seems to be difficult to reproduce the condition of double proteinopathy comprising TDP-43 and tau pathologies by altering one of molecules or genes shown above. This fact suggests that pathogeneses of TDP-43 proteinopathies and tauopathies arise from multifactorial and polygenetic processes. Further investigations to clarify the pathogenetic factors that are shared by a broad spectrum of neurodegenerative disorders will establish key therapeutic targets.


## Figures and Tables

**Figure 1 ijms-23-15755-f001:**
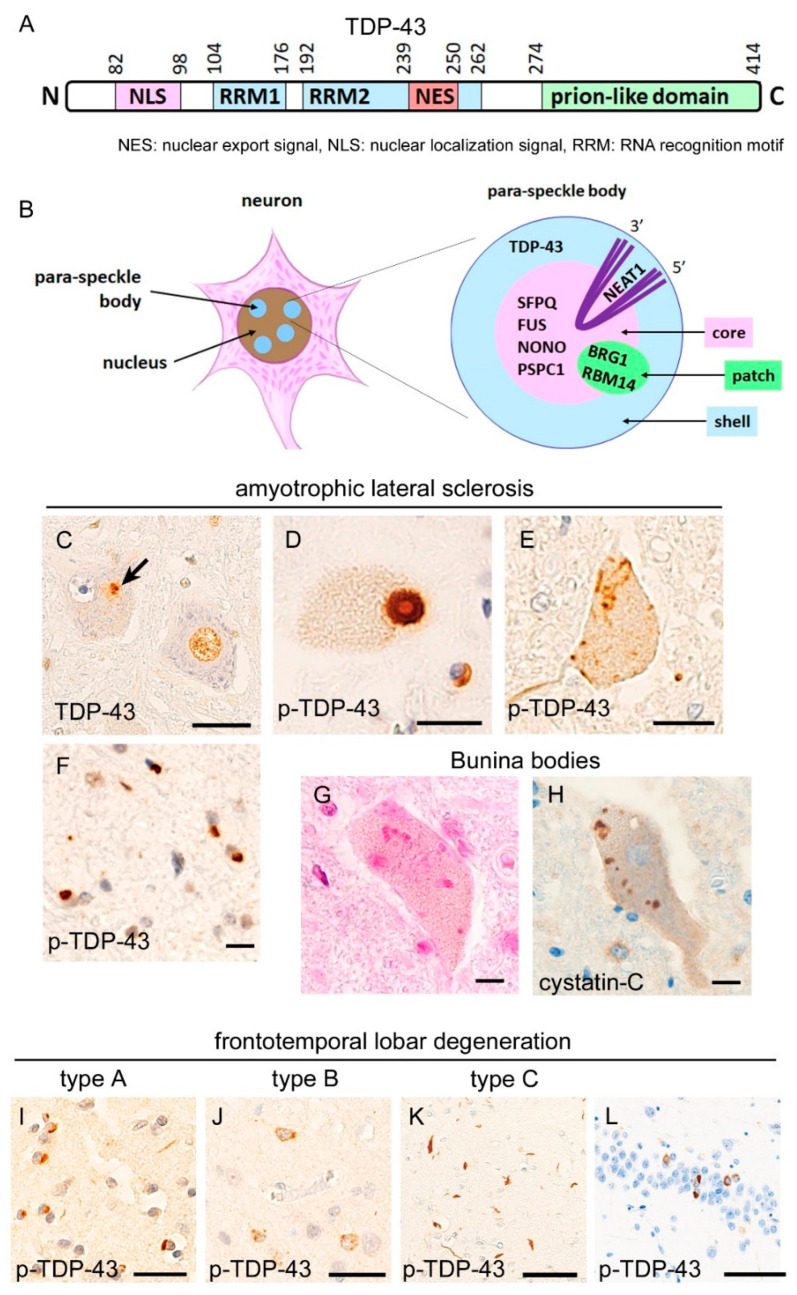
Neuropathological findings of TDP-43-proteinopathies. (**A**) TDP-43 contains nuclear localization (NLS) and export (NES) signals and RNA recognition motifs (RRMs). The C-terminal side has a glycine-rich domain (prion-like domain) [9,12]. (**B**) Intranuclear para-speckle body is an important scaffold of RNA-binding proteins, including TDP-43, SFPQ, and FUS, interacting with V-shaped lncRNA, NEAT1. The shell comprises the TDP-43 and 3′/5′-ends of NEAT1, whereas the core comprises SFPQ, FUS, NONO, and PSPC1 [14]. Panels (**C**–**H**) display pathological findings of spinal cord motor neurons from amyotrophic lateral sclerosis patients. TDP-43 pathology comprises mislocalization from nucleus and cytoplasmic aggregates (**C**, arrow). TDP-43 aggregates often appear as round (**D**) or skein-like (**E**) inclusions. Oligodendrocytic inclusions are also observed (**F**). Bunina bodies are eosinophilic granular inclusions (**G**), which are immunopositive for cystatin-C (**H**). Panels (**I**–**L**) display pathological findings of frontal cortex of frontotemporal lobar degeneration patients. Type A is characterized by small cytoplasmic inclusions and short and thin neuritic aggregates, predominantly in the superficial layers (**I**). Type B shows cytoplasmic aggregates throughout all cortical layers (**J**). Type C demonstrates thick and long neuritic aggregates, prominently in the superficial layers. Hippocampal granule cells are preferentially involved by FTLD-TDP (**L**). (**C**) Anti-TDP-43, (**H**) cystatin-C, and (**D**–**F**,**I**–**K**) phosphorylated TDP-43 immunohistochemistry. Scale bars: (**L**) 50 μm, (**C**,**I**–**K**), 20 μm, and (**D**–**H**) 10 μm. BRG1: brahma-related gene 1, FUS: fused-in-sarcoma, NONO: non-POU domain-containing octamer-binding protein, NEAT1: nuclear paraspeckle assembly transcript 1, PSPC1: paraspeckle component 1, RBG14: RNA binding motif protein 14, SFPQ: splicing factor proline and glutamine rich, and TDP-43: transactivation response DNA binding protein 43 kDa.

**Figure 2 ijms-23-15755-f002:**
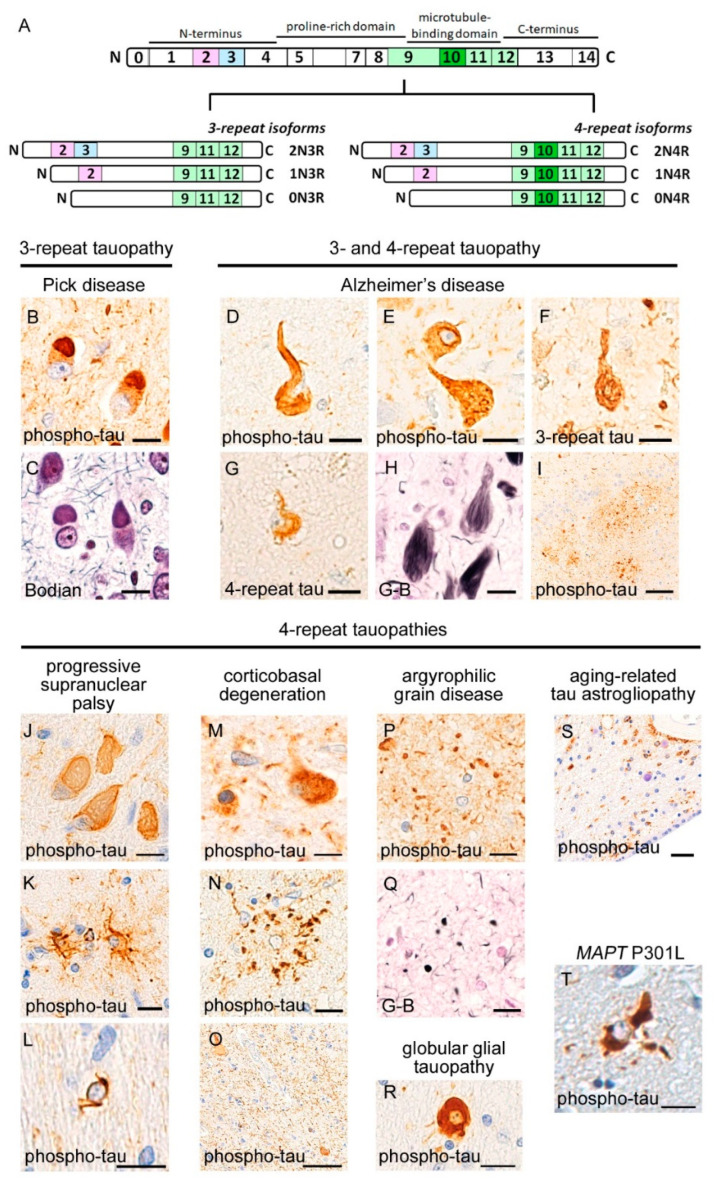
Neuropathological findings of tauopathies. Microtubule associated protein tau (MAPT) gene expressed in human brain is shown (**A**). Alternative splicing of exons 2, 3, and 10 generates six isoforms, which are subclassified into 3 repeat and 4 repeat isoforms by the absence or presence of exon 10, respectively. Pick disease is neuropathologically characterized by Pick bodies in the neuronal soma, which are densely labeled using anti-hyperphosphorylated tau immunohistochemistry (**B**) and silver staining methods (**C**). Alzheimer’s disease shows neurofibrillary tangles (**D**) or pre-tangles (**E**) in the neuronal soma. The neuronal inclusions contain 3 repeat (**F**) and 4 repeat (**G**) tau. Neurofibrillary tangles are densely labelled using silver staining methods (**H**). Senile plaques often exhibit tau epitopes (**I**), as well as Aβ-amyloid. Progressive supranuclear palsy is characterized by globose neurofibrillary tangles (**J**), tufted astrocytes (**K**), and oligodendrocytic coiled bodies (**L**). Corticobasal degeneration demonstrates pretangles (**M**) and astrocytic plaques (**N**). Dense neuropil threads of tau aggregations in the deep cortical layers and white matters are also characteristic (**O**). Argyrophilic grain disease shows “grains” spreading in the neuropil, which are labeled using anti-hyperphosphorylated tau immunohistochemistry (**P**) and silver staining methods (**Q**). Globular glial tauopathy prominently displays thick tau aggregates in the oligodendroglia (**R**). Aging-related tau astrogliopathy shows fuzzy or granular aggregates of tau, which are prominent in the subpial regions (**S**). *MAPT* P301L mutation, a common variant of the exon 10, demonstrates 4 repeat tauopathy with various morphologies; the panel shows thick astrocytic aggregation that differs from tufted astrocytes or astrocytic plaques (**T**). (**A**–**H**) Hippocampus, (**I**) midbrain tectum, (**J**–**N**) frontal lobe, (**O**,**P**) parahippocampal gyrus, (**Q**,**R**) temporal lobe, and (**S**) putamen. (**A**,**C**,**D**,**H**,**I**–**O**,**Q**–**S**) anti-phosphorylated tau (AT-8) immunohistochemistry, (**E**) anti-3-repeat tau (RD3) immunohistochemistry, (**F**) anti-4-repeat tau (RD4) immunohistochemistry, (**B**) Bodian staining, and (**G**,**P**) Gallyas-Braak (GB) staining. Scale bars: (**A**–**G**,**I**–**Q**,**S**) 10 μm and (**H**,**R**) 50 μm.

**Figure 4 ijms-23-15755-f004:**
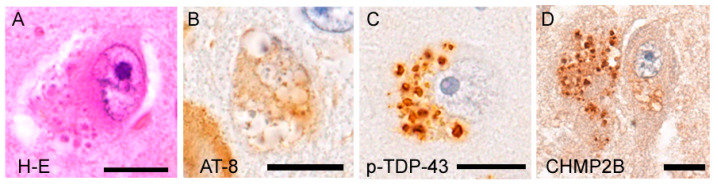
Granurovacuolar degeneration. Granurovacuolar degeneration (GVD) is characterized by a cluster of slightly basophilic granules within fused vacuoles (**A**). The granules are immunopositive for hyperphosphorylated tau (**B**), phosphorylated TDP-43 (**C**), and CHMP2B (**D**) [87,91]. Note that cytoplasm of the GVD-containing neuron also shows diffuse immunoreactivity to hyperphosphorylated tau (**B**). Bars = 10 μm. (**A**–**D**) Hippocampus.

**Figure 5 ijms-23-15755-f005:**
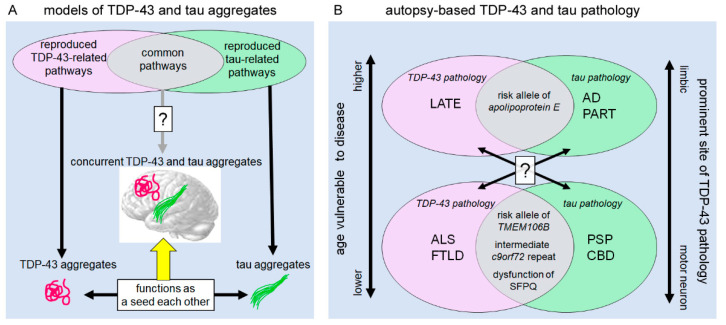
Overlapping of TDP-43 and tau pathologies in disease models and patients. (**A**) Although the upstream pathways driving TDP-43 or tau aggregates are hypothesized to overlap partially, the multiple proteinopathy comprising TDP-43 and tau pathology is unlikely to be reproduced by a single pathway in models. However, studies have shown that TDP-43 and tau aggregates have seed activities for each other, resulting in concurrent aggregations (a yellow arrow). (**B**) Autopsy-based research has suggested that the pathological findings of tauopathies and TDP-43 proteinopathies are partially overlapped and correlated with each other. Some studies have emphasized the possible links between AD and LATE or between PSP/CBD and ALS/FTLD-TDP, addressing shared pathogenic factors.

**Table 1 ijms-23-15755-t001:** Various conditions associated with comorbidity of combined TDP-43 and tau pathology.

Disorders	Pathogenesis/Key Molecule	Primary Aggregation	Description of Tau Pathology	Description of TDP-43 Pathology	Note, References
Aged people			PART/AD-NC	LATE	[4,7,8]
ALS/FTLD-TDP (sporadic or *GRN, c9orf72,* and *CHMP2B* mutations)		TDP-43	mostly PART/AD-NC, but unclassifiable in some cases	ALS/FTLD-TDP	[81,82,83,84,94]
ALS with *OPTN* E478G mutation	pathologic *OPTN* mutation	TDP-43	neuronal and glial 3R/4R-tau inclusions	ALS/FTLD-TDP	[106]
FTD with *FUS* Q140H mutation	*FUS* mutation with unclear significance	3R and 4R-tau	neuronal and glial 3R/4R-tau inclusions	granular neuronal inclusions in frontotemporal cortices	FUS pathology is absent [98]
ALS/FTLD-TDP with *TMEM106B* rs1990622 A/A	risk allele for TDP-43 pathology	TDP-43	neuronal and astrocytic 4R-tau aggregates	ALS/FTLD-TDP	[125]
Perry syndrome	pathogenic *DCTN1* mutation	TDP-43	PSP-like (tau isoforms are not described)	FTD-MND-like, with prominent involvement of substantia nigra	[104,107]
PSP		4R-tau	PSP-NC	prominent in hippocampus, ALS-like in motor neuron	[66,67,79,80]
CBD		4R-au	CBD-NC	prominent in midbrain, ALS-like in motor neuron	[68,69,79,80]
FTD with *MAPT* IVS10 + 16 mutation	pathogenic *MAPT* mutation	4R-tau	GGT	FTLD-TDP type A or B-like, densely spread	[111]
AD		3R and 4R-tau, Aβ-amyloid	AD-NC	LATE or FTLD-TDP type A like	[10,11,64,65]
Lewy body disease		α-synuclein	PART/AD-NC or colocalized with Lewy bodies	FTLD-TDP type A-like	[70,71,72]
Guam ALS/PDC		tau, TDP-43, α-synuclein	NFT, granular or thorn-shaped astrocytic aggregates	ALS-TDP-like	[113]
Kii peninsula ALS/PDC		tau, TDP-43, α-synuclein	NFT prominent in presubiculum	ALS-TDP-like	[114]
Neuronal ceroid lipofuscinosis, including homozygous *GRN* mutation	lysosomal storage disease		mostly PART/AD-NC, but unclassifiable in some cases	cytoplasm of ballooned neurons	[103,108,109]
Huntington disease	Huntingtin CAG repeat	CAG repeat	PART/AD-NC	colocalized with cytoplasmic/neuritic inclusions of huntingtin	[99,110]
NBIA type 1	pathogenic *PANK2* mutation		broadly extended NFT	neuronal and glial inclusions	[105]
periventricular nodular heterotopia			focal NFT	neuronal cytoplasmic and neuritic inclusions, thorn-shaped astrocytic aggregates, and a perivascular inclusion body	immunoreactivities were presented within heterotopia [112]
glioneuronal tumors	activated mTOR signaling		diffuse cytoplasmic inclusions or neuropil threads in tumor	GVD-like	[75]
chronic traumatic encephalopathy		3R and 4R-tau	neuronal and glial inclusions prominent nearby traumatic sites and periventricular regions	sparse neuronal inclusions out of traumatic sites	[74,97]
anti-IgLON5-related tauopathy	anti-IgLON5 antibody	3R and 4R-tau	3R/4R-tau pathology in hypothalamus and brainstem tegmentum	microglial and neuronal TDP-43 pathology in basal ganglia and midbrain	[102]
IBM (sporadic or *VCP* mutations)			muscle inclusion bodies	muscle inclusion bodies	GVD-related epitopes are also present [96,100,101]

Abbreviations: AD Alzheimer’s disease, ALS amyotrophic lateral sclerosis, CBD corticobasal degeneration, CHMP2B charged multivesicular body protein 2B, DCTN dynactin, FTD: behavior-variant frontotemporal dementia, FTD-MND FTD with motor neuron disease, FTLD frontotemporal lobar degeneration, FUS fused-in-sarcoma, GGT globular glial tauopathy, GVD granulovacuolar degeneration, IBM inclusion body myopathy, LATE limbic-predominant age-related TDP-43 encephalopathy, MAPT microtubule-associated protein tau, mTOR mammalian target of rapamycin, NBIA neurodegeneration with brain iron accumulation, NC neuropathological change, NFT neurofibrillary tangle, OPTN optineurin, PART primary age-related tauopathy, PDC parkinsonism–dementia complex, PSP progressive supranuclear palsy, TDP-43 transactivation response DNA binding protein 43 kDa, TMEM106B transmembrane protein 106B, VCP valocin-containing protein, 3R-tau 3-repeat tau, and 4R-tau 4-repeat tau.

## Data Availability

Not applicable.

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
