# Peer review of "TDP-43 Proteinopathy and Tauopathy: Do They Have Pathomechanistic Links?"

_ijms, 2022, doi:10.3390/ijms232415755_

Round 1
Reviewer 1 Report
This review describes in a detail manner the potential link between TDP-43 and Tau. For the most part, the paper is well written and very comprehensible, but there are few things that need to be addressed before publication:
1. The section on “Neuropathology of tau” is quite short. It should be expanded considering that there are lots of reports on the role of Tau in the major neurodegenerative disease.
2. Add a picture to show Tau pathology in AD, PICK, and FTLD.
3. There are typos throughout the paper.
1
Author Response
Dear Reviewer:
We deeply appreciate deep insights about manuscript review from you. We have revised the paper on the basis of reviewers’ comments. Our reply to each comment is shown below.
- The section on “Neuropathology of tau” is quite short. It should be expanded considering that there are lots of reports on the role of Tau in the major neurodegenerative disease.
Answer: We agree with this point. We have added more explanation about neuropathological findings of tauopathies and tau-related pathway focusing on its misfolding.
- Add a picture to show Tau pathology in AD, PICK, and FTLD.
Answer: We appreciate this suggestion. We have added figure panels for this section.
- There are typos throughout the paper.
Answer: We have re-edited writing throughout the paper.
Reviewer 2 Report
The question of the relationship between TDP-43 proteinopathies and tauopathies is of interest to a wide audience, however, in its present form, the review is neither interesting nor useful. Because there is practically no analysis of the collected information.
But I want to start my remarks from a purely formal point! The text (including the title and abstract) is filled with undeciphered abbreviations. Due to the specifics of the field, it is necessary to introduce a table of abbreviations.
Further, I would suggest starting each section from the beginning, for example:
1) Section 2. Neuropathology of TDP-43 proteinopathies.
should begin with a description of the TAR DNA-binding protein 43 protein and only then move on to its role in neuropathology, focusing on mutations, the nature of TDP-43-containing aggregates and various ways of their detection.
Some tables listing the pathologies associated with the appearance of TDP-43-containing aggregates in various pathologies would be very useful.
2) A similar approach is desirable when describing Tau and Tauopathies
3) Tables or diagrams (or other figures) illustrating the joint occurrence of TDP-43- and tau-containing aggregates should be present in the respective charter with description in what cases and by what method it was established.
4) Why Aging-related TDP-43 and tau pathology was separated in the separate charter should be explained
5) The charter Interpretation of comorbid TDP-43 pathology does not provide any interpretation.
In the beginning the 3 theory was mentioned. Each of the theories should be discussed and the opinion of the authors, based on the analysis of the literature, should be expressed.
6) The last charter Do TDP-43 proteinopathies and tauopathies have mechanistic links? Is the most important. In it, again, I would like to see a table (s) or diagrams that list the currently known genetic (mutations) or protein (expression levels) factors (including the information how and where it was detected) associated with the appearance of the TDP-43 and tau protein aggregates together or separately, and perhaps some theories why certain aggregates accumulate in cells.
Author Response
Dear Reviewer:
We deeply appreciate deep insights about manuscript review from you. We have revised the paper on the basis of reviewers’ comments. Our reply to each comment is shown below.
-The text (including the title and abstract) is filled with undeciphered abbreviations. Due to the specifics of the field, it is necessary to introduce a table of abbreviations.
Answer: We appreciate this suggestion and have added a list of abbreviations in the head of manuscript.
-Section 2. Neuropathology of TDP-43 proteinopathies.
should begin with a description of the TAR DNA-binding protein 43 protein and only then move on to its role in neuropathology, focusing on mutations, the nature of TDP-43-containing aggregates and various ways of their detection.
Some tables listing the pathologies associated with the appearance of TDP-43-containing aggregates in various pathologies would be very useful.
Answer: We have added some physiological facts about TDP-43 in this section and figure panels explaining TDP-43 pathology. For making a table, please see a later comment.
-A similar approach is desirable when describing Tau and Tauopathies
Answer: Reviewer 1 also addressed this point. We have added some writing about tau protein, MAPT mutation, and tau pathology, showing figure panels.
- Tables or diagrams (or other figures) illustrating the joint occurrence of TDP-43- and tau-containing aggregates should be present in the respective charter with description in what cases and by what method it was established.
Answer: In the former version, illustrating of comorbid pathologies were shown with fragmented figures and resulted in low readability. During a revision, we unified those to correspond with the chapter ‘comorbid TDP-43 pathology in tauopathies and comorbid tau pathology in TDP-43 proteinopathies’.
Why Aging-related TDP-43 and tau pathology was separated in the separate charter should be explained.
Answer: In the revised version, the chapter of aging pathology was unified into ‘Neuropathology of TDP-43 proteinopathies’ and ‘Neuropathology of tauopathies’.
The charter Interpretation of comorbid TDP-43 pathology does not provide any interpretation.
Answer: In the revised version, interpretation of comorbid pathology was included in the chapter ‘comorbid TDP-43 pathology in tauopathies and comorbid tau pathology in TDP-43 proteinopathies’ with appropriate literatures.
The last charter Do TDP-43 proteinopathies and tauopathies have mechanistic links? Is the most important. In it, again, I would like to see a table (s) or diagrams that list the currently known genetic (mutations) or protein (expression levels) factors (including the information how and where it was detected) associated with the appearance of the TDP-43 and tau protein aggregates together or separately, and perhaps some theories why certain aggregates accumulate in cells.
Answer: We appreciate this suggestion. For autopsy (patient) level, the last chapter in the revised manuscript contains a table describing conditions in which concomitant TDP-43 and tau pathologies have been reported, showing literatures. For basic research (model) level, we have addressed some studies successfully reproducing TDP-43 or tau pathology. Numerous basic studies about each TDP-43 or tau aggregation exist, hence we introduced those addressing certain molecules/genes responsible for both TDP-43 and tau aggregates in this section.
Round 2
Reviewer 2 Report
The authors have significantly improved the quality of their review. But when the main shortcomings were eliminated, it became noticeable that the chapter introduction does not carry any semantic load. It would probably be worth expanding this part of the manuscript, explaining what the main purpose of this review is, and how it can help in research aimed at combating the neurological diseases in question.
Author Response
Reviewer comment: The authors have significantly improved the quality of their review. But when the main shortcomings were eliminated, it became noticeable that the chapter introduction does not carry any semantic load. It would probably be worth expanding this part of the manuscript, explaining what the main purpose of this review is, and how it can help in research aimed at combating the neurological diseases in question.
Answer: We appreciate referee’s suggestion about the introduction. We added some writings to explain the agenda of paper in the introduction.